# PerforatedCNNs: Acceleration through Elimination of Redundant Convolutions

**Michael Figurnov**[1,2], **Aijan Ibraimova**[4], **Dmitry Vetrov**[1,3], and **Pushmeet Kohli**[5]

[1]National Research University Higher School of Economics  [2]Lomonosov Moscow State University
[3]Yandex  [4]Skolkovo Institute of Science and Technology  [5]Microsoft Research
michael@figurnov.ru, aijan.ibraimova@gmail.com, vetrovd@yandex.ru,
pkohli@microsoft.com

## Abstract

We propose a novel approach to reduce the computational cost of evaluation of convolutional neural networks, a factor that has hindered their deployment in low-power devices such as mobile phones. Inspired by the loop perforation technique from source code optimization, we speed up the bottleneck convolutional layers by skipping their evaluation in some of the spatial positions. We propose and analyze several strategies of choosing these positions. We demonstrate that perforation can accelerate modern convolutional networks such as AlexNet and VGG-16 by a factor of $2\times$ - $4\times$. Additionally, we show that perforation is complementary to the recently proposed acceleration method of Zhang et al. [28].

## 1 Introduction

The last few years have seen convolutional neural networks (CNNs) emerge as an indispensable tool for computer vision. However, modern CNNs have a high computational cost of evaluation, with convolutional layers usually taking up over 80% of the time. For instance, VGG-16 network [25] for the problem of object recognition requires $1.5 \cdot 10^{10}$ floating point multiplications per image. These computational requirements hinder the deployment of such networks on systems without GPUs and in scenarios where power consumption is a major concern, such as mobile devices.

The problem of trading accuracy of computations for speed is well-known within the software engineering community. One of the most prominent methods for this problem is *loop perforation* [18, 19, 24]. In a nutshell, this technique isolates loops in the code that are not critical for the execution, and then reduces their computational cost by skipping some iterations. More recently, researchers have considered problem-dependent perforation strategies that exploit the structure of the problem [23].

Inspired by the general principle of perforation, we propose to reduce the computational cost of CNN evaluation by exploiting the spatial redundancy of the network. Modern CNNs, such as AlexNet, exploit this redundancy through the use of strides in the convolutional layers. However, using the convolutional strides changes the architecture of the network (intermediate representations size and the number of weights in the first fully-connected layer), which might be undesirable. Instead of using strides, we argue for the use of interpolation (perforation) of responses in the convolutional layer. A key element of this approach is the choice of the perforation mask, which defines the output positions to evaluate exactly. We propose several approaches to select the perforation masks and a method of choosing a combination of perforation masks for different layers. To restore the network accuracy, we perform fine-tuning of the perforated network. Our experiments show that this method can reduce the evaluation time of modern CNN architectures proposed in the literature by a factor of $2\times$ - $4\times$ with a small decrease in accuracy.

## 2 Related Work

Reducing the computational cost of CNN evaluation is an active area of research, with both highly optimized implementations and approximate methods investigated.

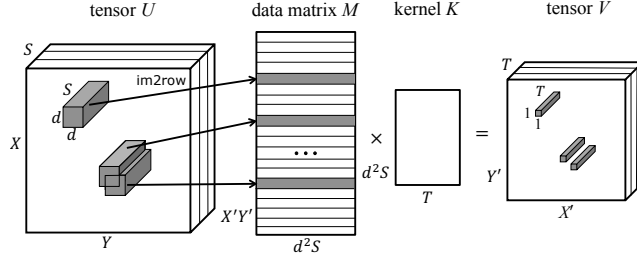

Figure 1: Reduction of convolutional layer evaluation to matrix multiplication. Our idea is to leave only a subset of rows (defined by a *perforation mask*) in the data matrix $M$ and to interpolate the missing output values.

Implementations that exploit the parallelism available in computational architectures like GPUs (cuda-convnet2 [13], CuDNN [3]) have allowed to significantly reduce the evaluation time of CNNs. Since CuDNN internally reduces the computation of convolutional layers to the matrix-by-matrix multiplication (without explicitly materializing the data matrix), our approach can potentially be incorporated into this library. In a similar vein, the use of FPFGAs [22] leads to better trade-offs between speed and power consumption. Several papers [5, 9] showed that CNNs may be efficiently evaluated using low precision arithmetic, which is important for FPFGA implementations. Most approximate methods of decreasing the CNN computational cost exploit the redundancies of the convolutional kernel using low-rank tensor decompositions [6, 10, 16, 28]. In most cases, a convolutional layer is replaced by several convolutional layers applied sequentially, which have a much lower total computational cost. We show that the combination of perforation with the method of Zhang et al. [28] improves upon both approaches.

For spatially sparse inputs, it is possible to exploit this sparsity to speed up evaluation and training [8]. While this approach is similar to ours in the spirit, we do not rely on spatially sparse inputs. Instead, we sparsely sample the outputs of a convolutional layer and interpolate the remaining values.

In a recent work, Lebedev and Lempitsky [15] also decrease the CNN computational cost by reducing the size of the data matrix. The difference is that their approach reduces the convolutional kernel's support while our approach decreases the number of spatial positions in which the convolutions are evaluated. The two methods are complementary.

Several papers have demonstrated that it is possible to compress the parameters of the fully-connected layers (where most CNN parameters reside) with a marginal error increase [4, 21, 27]. Since our method does not directly modify the fully-connected layers, it is possible to combine these methods with our approach and obtain a fast and small CNN.

## 3 PerforatedCNNs

The section provides a detailed description of our approach. Before proceeding further, we introduce the notation that will be used in the rest of the paper.

**Notation.** A convolutional layer takes as input a tensor $U$ of size $X \times Y \times S$ and outputs a tensor $V$ of size $X' \times Y' \times T$, $X' = X - d + 1$, $Y' = Y - d + 1$. The first two dimensions are spatial (height and width), and the third dimension is the number of channels (for example, for an RGB input image $S = 3$). The set of $T$ convolution kernels $K$ is given by a tensor of size $d \times d \times S \times T$. For simplicity of notation, we assume unit stride, no zero-padding and skip the biases. The convolutional layer output may be defined as follows:

$$V(x, y, t) = \sum_{i=1}^{d} \sum_{j=1}^{d} \sum_{s=1}^{S} K(i, j, s, t) U(x + i - 1, y + j - 1, s) \tag{1}$$

Additionally, we define the set of all spatial indices (positions) of the output $\Omega = \{1, \ldots, X'\} \times \{1, \ldots, Y'\}$. Perforation mask $I \subseteq \Omega$ is the set of indices in which the outputs are calculated exactly. Denote $N = |I|$ the number of positions to be calculated exactly, and $r = 1 - \frac{N}{|\Omega|}$ the *perforation rate*.

**Reduction to matrix multiplication.** To achieve high computational performance, many deep learning frameworks, including Caffe [12] and MatConvNet [26], reduce the computation of convolutional

layers to the heavily-optimized matrix-by-matrix multiplication routine of basic linear algebra packages. This process, sometimes referred to as *lowering*, is illustrated in fig. 1. First, a *data matrix $M$* of size $X'Y' \times d^2 S$ is constructed using `im2row` function. The rows of $M$ are elements of patches of input tensor $U$ of size $d \times d \times S$. Then, $M$ is multiplied by the kernel tensor $K$ reshaped into size $d^2 S \times T$. The resulting matrix of size $X'Y' \times T$ is the output tensor $V$, up to a reshape. For a more detailed exposition, see [26].

## 3.1 Perforated convolutional layer

In this section we present the *perforated convolutional layer*. In a small fraction of spatial positions, the outputs of the proposed layer are equal to the outputs of a usual convolutional layer. The remaining values are interpolated using the nearest neighbor from this set of positions. We evaluate other interpolation strategies in appendix A.

The perforated convolutional layer is a generalization of the standard convolutional layer. When the perforation mask is equal to all the output spatial positions, the perforated convolutional layer's output equals the conventional convolutional layer's output.

Formally, let $I \subseteq \Omega$ be the *perforation mask* of spatial output to be calculated exactly (the constraint that the masks are shared for all channels of the output is required for the reduction to matrix multiplication). The function $\ell(x, y) : \Omega \to I$ returns the index of the nearest neighbor in $I$ according to Euclidean distance (with ties broken randomly):

$$\ell(x,y) = (\ell_1(x,y), \ell_2(x,y)) = \arg\min_{(x',y') \in I} \sqrt{(x-x')^2 + (y-y')^2}. \tag{2}$$

Note that the function $\ell(x, y)$ may be calculated in advance and cached.

The perforated convolutional layer output $\hat{V}$ is defined as follows:

$$\hat{V}(x,y,t) = V(\ell_1(x,y), \ell_2(x,y), t), \tag{3}$$

where $V(x, y, t)$ is the output of the usual convolutional layer, defined by (1). Since $\ell(x, y) = (x, y)$ for $(x, y) \in I$, the outputs in the spatial positions $I$ are calculated exactly. The values in other positions are interpolated using the value of the nearest neighbor. To evaluate a perforated convolutional layer, we only need to calculate the values $V(x, y, t)$ for $(x, y) \in I$, which can be done efficiently by reduction to matrix multiplication. In this case, the data matrix $M$ contains just $N = |I|$ rows, instead of the original $X'Y' = |\Omega|$ rows. Perforation is not limited to this implementation of a convolutional layer, and can be combined with other implementations that support strided convolutions, such as the direct convolution approach of cuda-convnet2 [13].

In our implementation, we only store the output values $V(x, y, t)$ for $(x, y) \in I$. The interpolation is performed implicitly by masking the reads of the following pooling or convolutional layer. For example, when accelerating conv3 layer of AlexNet, the interpolation cost is transferred to conv4 layer. We observe no slowdown of the conv4 layer when using GPU, and a 0-3% slowdown when using CPU. This design choice has several advantages. Firstly, the memory size required to store the activations is reduced by a factor of $\frac{1}{1-r}$. Secondly, the following non-linearity layers and $1 \times 1$ convolutional layers are also sped up since they are applied to a smaller number of elements.

## 3.2 Perforation masks

We propose several ways of generating the perforation masks, or choosing $N$ points from $\Omega$. We visualize the perforation masks $I$ as binary matrices with black squares in the positions of the set $I$. We only consider the perforation masks that are independent of the input object and leave exploration of input-dependent perforation masks to the future work.

**Uniform** perforation mask is just $N$ points chosen randomly without replacement from the set $\Omega$. However, as can be seen from fig. 2a, for $N \ll |\Omega|$, the points tend to cluster. This is undesirable because a more scattered set $I$ would reduce the average distance to the set $I$.

**Grid** perforation mask is a set of points $I = \{a(1), \ldots, a(K_x)\} \times \{b(1), \ldots, b(K_y)\}$, see fig. 2b. We choose the values of $a(i), b(i)$ using the *pseudorandom integer sequence generation* scheme of [7].

**Pooling structure** mask exploits the structure of the overlaps of pooling operators. Denote by $A(x, y)$ the number of times an output of the convolutional layer is used in the pooling operators. The grid-like pattern as in fig. 2d is caused by a pooling of size $3 \times 3$ with stride 2 (such parameters are used e.g. in Network in Network and AlexNet). The pooling structure mask is obtained by picking top-$N$ positions with the highest values of $A(x, y)$, with ties broken randomly, see fig. 2c.

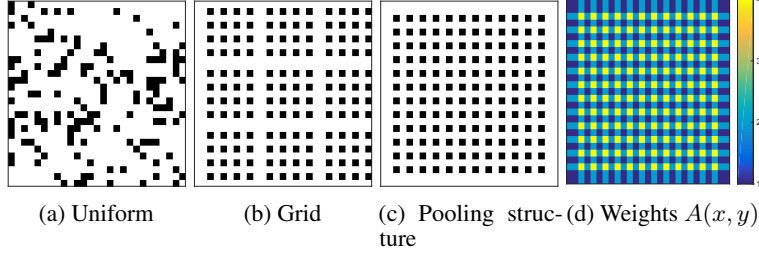

| (a) Uniform | (b) Grid | (c) Pooling struc-<br>ture | (d) Weights $A(x, y)$ |
|---|---|---|---|

Figure 2: Perforation masks, AlexNet conv2, $r = 80.25\%$. Best viewed in color.

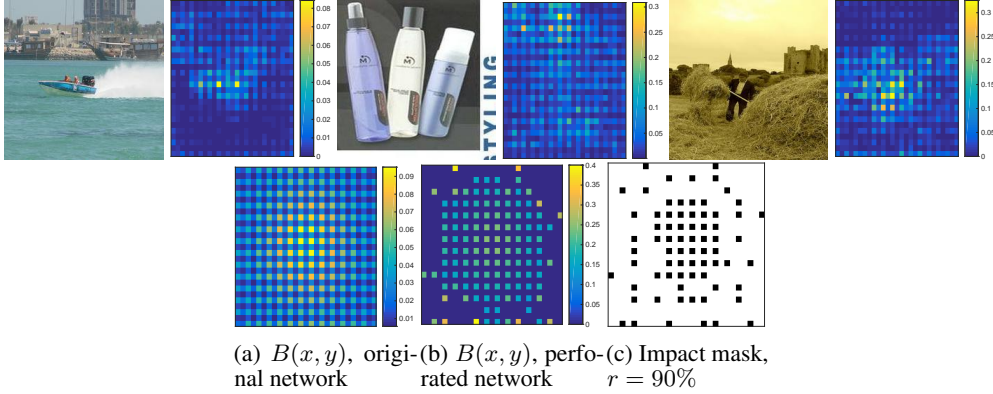

(a) $B(x, y)$, origi-(b) $B(x, y)$, perfo-(c) Impact mask,
nal network      rated network      $r = 90\%$

Figure 3: **Top:** ImageNet images and corresponding values of impact $G(x, y; V)$ for AlexNet conv2.
**Bottom:** average impacts and impact perforation mask for AlexNet conv2. Best viewed in color.

**Impact** mask estimates the impact of perforation of each position on the CNN loss function, and
then removes the least important positions. Denote by $L(V)$ the loss function of the CNN (such as
negative log-likelihood) as a function of the considered convolutional layer outputs $V$. Next, suppose
$V'$ is obtained from $V$ by replacing one element $(x_0, y_0, t_0)$ with a neutral value zero. We estimate
the *impact* of a position as a first-order Taylor approximation of the magnitude of change of $L(V)$:

$$|L(V') - L(V)| \approx \Big| \sum_{x=1}^{X} \sum_{y=1}^{Y} \sum_{t=1}^{T} \frac{\partial L(V)}{\partial V(x, y, t)} (V'(x, y, t) - V(x, y, t)) \Big|$$

$$= \Big| \frac{\partial L(V)}{\partial V(x_0, y_0, t_0)} V(x_0, y_0, t_0) \Big|. \tag{4}$$

The value $\frac{\partial L(V)}{\partial V(x_0, y_0, t_0)}$ may be obtained using backpropagation. In the case of a perforated convolu-
tional layer, we calculate the derivatives with respect to the convolutional layer output $V$ (not the
interpolated output $\hat{V}$). This makes the impact of the previously perforated positions zero and sums
the impact of the non-perforated positions over all the outputs which share the value.

Since we are interested in the total impact of a spatial position $(x, y) \in \Omega$, we take a sum over all the
channels and average this estimate of impacts over the training dataset:

$$G(x, y; V) = \sum_{t=1}^{T} \Big| \frac{\partial L(V)}{\partial V(x, y, t)} V(x, y, t) \Big| \tag{5}$$

$$B(x, y) = \mathbb{E}_{V \sim \text{training set}} G(x, y; V) \tag{6}$$

Finally, the impact mask is formed by taking the top-$N$ positions with the highest values of $B(x, y)$.
Examples of the values of $G(x, y; V)$, $B(x, y)$ and impact mask are shown on fig. 3. Note that the
regions of the high value of $G(x, y; V)$ usually contain the most salient features of the image. The
averaged weights $B(x, y)$ tend to be higher in the center since ImageNet's images usually contain a
centered object. Additionally, a grid-like structure of pooling structure mask is automatically inferred.

| Network | Dataset | Error | CPU time | GPU time | Mem. | Mult. | # conv |
|---------|---------|-------|----------|----------|------|-------|--------|
| NIN | CIFAR-10 | top-1 10.4% | 4.6 ms | 0.8 ms | 5.1 MB | $2.2 \cdot 10^8$ | 3 |
| AlexNet | ImageNet | top-5 19.6% | 16.7 ms | 2.0 ms | 6.6 MB | $0.5 \cdot 10^9$ | 5 |
| VGG-16 | | top-5 10.1% | 300 ms | 29 ms | 110 MB | $1.5 \cdot 10^{10}$ | 13 |

Table 1: Details of the CNNs used for the experimental evaluation. Timings, memory consumption and number of multiplications are normalized by the batch size. Memory consumption is the memory required to store activations (intermediate results) of the network during the forward pass.

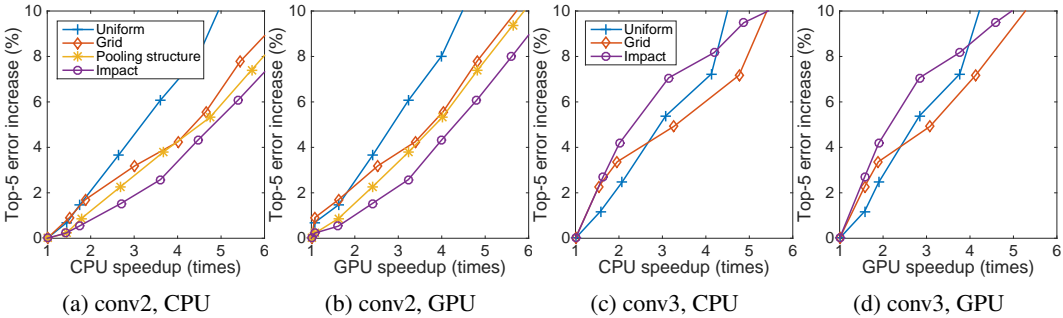

(a) conv2, CPU  (b) conv2, GPU  (c) conv3, CPU  (d) conv3, GPU

Figure 4: Acceleration of a single layer of AlexNet for different mask types without fine-tuning. Values are averaged over 5 runs.

Since perforation of a layer changes the impacts of all the layers, in the experiments we iterate between increasing the perforation rate of a layer and recalculation of impacts. We find that this improves results by co-adapting the perforation masks of different convolutional layers.

### 3.3 Choosing the perforation configurations

For whole network acceleration, it is important to find a combination of per-layer perforation rates that would achieve high speedup with low error increase. To do this, we employ a simple greedy strategy. We use a single perforation mask type and a fixed range of increasing perforation rates. Denote by $t$ the evaluation time of the accelerated network and by $e$ the objective (we use negative log-likelihood for a subset of training images). Let $t_0$ and $e_0$ be the respective values for the non-accelerated network. At each iteration, we try to increase the perforation rate for each layer and choose the layer for which this results in the minimal value of the cost function $\frac{e-e_0}{t_0-t}$.

## 4 Experiments

We use three convolutional neural networks of increasing size and computational complexity: Network in Network [17], AlexNet [14] and VGG-16 [25], see table 1. In all networks, we attempt to perforate all the convolutional layers, except for the $1 \times 1$ convolutional layers of NIN. We perform timings on a computer with a quad-core Intel Core i5-4460 CPU, 16 GB RAM and a nVidia Geforce GTX 980 GPU. The batch size used for timings is 128 for NIN, 256 for AlexNet and 16 for VGG-16. The networks are obtained from Caffe Model Zoo. For AlexNet, the Caffe reimplementation is used which is slightly different from the original architecture (pooling and normalization layers are swapped). We use a fork of MatConvNet framework for all experiments, except for fine-tuning of AlexNet and VGG-16, for which we use a fork of Caffe. The source code is available at `https://github.com/mfigurnov/perforated-cnn-matconvnet`, `https://github.com/mfigurnov/perforated-cnn-caffe`.

We begin our experiments by comparing the proposed perforation masks in a common benchmark setting: acceleration of a single AlexNet layer. Then, we compare whole-network acceleration with the best-performing masks to baselines such as decrease of input images size and an increase of strides. We proceed to show that perforation scales to large networks by presenting the whole-network acceleration results for AlexNet and VGG-16. Finally, we demonstrate that perforation is complementary to the recently proposed acceleration method of Zhang et al. [28].

| Method | CPU time ↓ | Error ↑ (%) |
|---|---|---|
| Impact, $r = \frac{3}{4}$, $3 \times 3$ filters | $9.1\times$ | $+1$ |
| Impact, $r = \frac{5}{6}$ | $5.3\times$ | $+1.4$ |
| Impact, $r = \frac{4}{5}$ | $4.2\times$ | $+0.9$ |
| Lebedev and Lempitsky [15] | $20\times$ | top-1 $+1.1$ |
| Lebedev and Lempitsky [15] | $9\times$ | top-1 $+0.3$ |
| Jaderberg et al. [10] | $6.6\times$ | $+1$ |
| Lebedev et al. [16] | $4.5\times$ | $+1$ |
| Denton et al. [6] | $2.7\times$ | $+1$ |

Table 2: Acceleration of AlexNet's conv2. **Top:** our results after fine-tuning, **bottom:** previously published results. Result of [10] provided by [16]. The experiment with reduced spatial size of the kernel ($3 \times 3$, instead of $5 \times 5$) suggests that perforation is complementary to the "brain damage" method of [15] which also reduces the spatial support of the kernel.

### 4.1 Single layer results

We explore the speedup-error trade-off of the proposed perforation masks on the two bottleneck convolutional layers of AlexNet, conv2 and conv3, see fig. 4. The pooling structure perforation mask is only applicable to the conv2 because it is directly followed by a max-pooling, whereas the conv3 is followed by another convolutional layer. We see that impact perforation mask works best for the conv2 layer while grid mask performs very well for conv3. The standard deviation of results is small for all the perforation masks, except the uniform mask for high speedups (where the grid mask outperforms it). The results are similar for both CPU and GPU, showing the applicability of our method for both platforms. Note that if we consider the best perforation mask for each speedup value, then we see that the conv2 layer is easier to accelerate than the conv3 layer. We observe this pattern in other experiments: layers immediately followed by a max-pooling are easier to accelerate than the layers followed by a convolutional layer. Additional results for NIN network are presented in appendix B.

We compare our results after fine-tuning to the previously published results on the acceleration of AlexNet's conv2 in table 2. Motivated by the results of [15] that the spatial support of conv2 convolutional kernel may be reduced with a small error increase, we reduce the kernel's spatial size from $5 \times 5$ to $3 \times 3$ and apply the impact perforation mask. This leads to the $9.1\times$ acceleration for $1\%$ top-5 error increase. Using the more sophisticated method of [15] to reduce the spatial support may lead to further improvements.

### 4.2 Baselines

We compare PerforatedCNNs with the baseline methods of decreasing the computational cost of CNNs by exploiting the spatial redundancy. Unlike perforation, these methods decrease the size of the activations (intermediate outputs) of the CNN. For a network with fully-connected (FC) layers, this would change the number of CNN parameters in the first FC layer, effectively modifying the architecture. To avoid this, we use CIFAR-10 NIN network, which replaces FC layers with global average pooling (mean-pooling over all spatial positions in the last layer).

We consider the following baseline methods. **Resize**. The input image is downscaled with the aspect ratio preserved. **Stride**. The strides of the convolutional layers are increased, making the activations spatially smaller. **Fractional stride**. Motivated by fractional max-pooling [7], we introduce a more flexible modification of strides which evaluates convolutions on a non-regular grid (with a varying step size), providing a more fine-grained control over the activations size and speedup. We use grid perforation mask generation scheme to choose the output positions to evaluate.

We compare these strategies to perforation of all the layers with the two types of masks which performed best in the previous section: **grid** and **impact**. Note that "grid" is, in fact, equivalent to fractional strides, but with missing values being interpolated.

All the methods, except resize, require a parameter value per convolutional layer, leading to a large number of possible configurations. We use the original network to explore this space of configurations. For impact, we use the greedy algorithm. For stride, we evaluate all possible combinations of parameters. For grid and fractional strides, for each layer we consider the set of rates $\frac{1}{3}, \frac{1}{2}, \ldots, \frac{8}{9}, \frac{9}{10}$ (for fractional strides this is the fraction of convolutions calculated), and evaluate all combinations of such rates. Then, for each method, we build a Pareto-optimal front of parameters

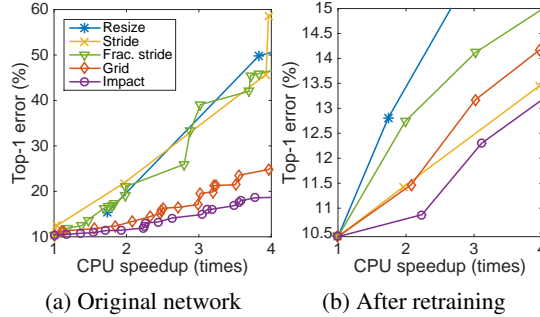

| (a) Original network | (b) After retraining |

Figure 5: Comparison of whole network perforation (grid and impact mask) with baseline strategies (resizing the input images, increasing the strides of convolutional layers) for acceleration of CIFAR-10 NIN network.

which produced smallest error increase for a given CPU speedup. Finally, we train the network weights "from scratch" (starting from a random initialization) for the Pareto-optimal configurations with accelerations close to $2\times, 3\times, 4\times$. For fractional strides, we use fine-tuning, since it performs significantly better than training from scratch.

The results are displayed on fig. 5. Impact perforation is the best strategy both for the original network and after training the network from scratch. Grid perforation is slightly worse. Convolutional strides are used in many CNNs, such as AlexNet, to decrease the computational cost of training and evaluation. Our results show that if changing the intermediate representations size and training the network from scratch is an option, then it is indeed a good strategy. Although more general, fractional strides perform poorly compared to strides, most likely because they "downsample" the outputs of a convolutional layer non-uniformly, making them hard to process by the next convolutional layer.

## 4.3 Whole network results

We evaluate the effect of perforation of all the convolutional layers of three CNN models. To tune the perforation rates, we employ the greedy method described in section 3.3. We use twenty perforation rates: $\frac{1}{3}, \frac{1}{2}, \frac{2}{3}, \dots, \frac{18}{19}, \frac{19}{20}$. For NIN and AlexNet we use the impact perforation mask. For VGG-16 we use the grid perforation mask as we find that it considerably simplifies fine-tuning. Using more than one type of perforation masks does not improve the results. Obtaining the perforation rates configuration takes about one day for the largest network we considered, VGG-16. In order to decrease the error of the accelerated network, we tune the network's weights. We do not observe any problems with backpropagation, such as exploding/vanishing gradients. The results are presented in table 3. Perforation damages the network performance significantly, but network weights tuning restores most of the accuracy. All the considered networks may be accelerated by a factor of two on both CPU and GPU, with under $2.6\%$ increase of error. Theoretical speedups (reduction of the number of multiplications) are usually close to the empirical ones. Additionally, the memory required to store network activations is significantly reduced by storing only the non-perforated output values.

## 4.4 Combining acceleration methods

A promising way to achieve high speedups with low error increase is to combine multiple acceleration methods. For this to succeed, the methods should exploit different *types* of redundancy in the network. In this section, we verify that perforation can be combined with the inter-channel redundancy elimination approach of [28] to achieve improved speedup-error ratios.

We reimplement the *linear asymmetric* method of [28]. It decomposes a convolutional layer with a $(d \times d \times S \times T)$ kernel (height-width-input channels-output channels) into a sequence of two layers, $(d \times d \times S \times T') \rightarrow (1 \times 1 \times T' \times T)$, $T' < T$. The second layer is typically very fast, so the overall speedup is roughly $\frac{T}{T'}$. When decomposing a perforated convolutional layer, we transfer the perforation mask to the first obtained layer.

We first apply perforation to the network and fine-tune it, as in the previous section. Then, we apply the inter-channel redundancy elimination method to this network. Finally, we perform the second round of fine-tuning with a much lower learning rate of 1e-9, due to exploding gradients. All the methods are tested at the theoretical speedup level of $4\times$. When the two methods are combined, the acceleration rate for each method is taken to be roughly equal. The results are presented in the table

| Network | Device | Speedup | Mult. ↓ | Mem. ↓ | Error ↑ (%) | Tuned error ↑ (%) |
|---|---|---|---|---|---|---|
| NIN | CPU | 2.2× | 2.5× | 2.0× | +1.5 | +0.4 |
| | | 3.1× | 4.4× | 3.5× | +5.5 | +1.9 |
| | | 4.2× | 6.6× | 4.4× | +8.3 | +2.9 |
| | GPU | 2.1× | 3.6× | 3.3× | +4.5 | +1.6 |
| | | 3.0× | 10.1× | 5.7× | +18.2 | +5.6 |
| | | 3.5× | 19.1× | 9.2× | +37.4 | +12.4 |
| AlexNet | CPU | 2.0× | 2.1× | 1.8× | +10.7 | +2.3 |
| | | 3.0× | 3.5× | 2.6× | +28.0 | +6.1 |
| | | 3.6× | 4.4× | 2.9× | +60.7 | +9.9 |
| | GPU | 2.0× | 2.0× | 1.7× | +8.5 | +2.0 |
| | | 3.0× | 2.6× | 2.0× | +16.4 | +3.2 |
| | | 4.1× | 3.4× | 2.4× | +28.1 | +6.2 |
| VGG-16 | CPU | 2.0× | 1.8× | 1.5× | +15.6 | +1.1 |
| | | 3.0× | 2.9× | 1.8× | +54.3 | +3.7 |
| | | 4.0× | 4.0× | 2.5× | +71.6 | +5.5 |
| | GPU | 2.0× | 1.9× | 1.7× | +23.1 | +2.5 |
| | | 3.0× | 2.8× | 2.4× | +65.0 | +6.8 |
| | | 4.0× | 4.7× | 3.4× | +76.5 | +7.3 |

Table 3: Full network acceleration results. Arrows indicate increase or decrease in the metric. Speedup is the wall-clock acceleration. Mult. is a reduction of the number of multiplications in convolutional layers (theoretical speedup). Mem. is a reduction of memory required to store the network activations. Tuned error is the error after training from scratch (NIN) or fine-tuning (AlexNet, VGG16) of the accelerated network's weights.

| Perforation | Asymm. [28] | Mult. ↓ | Mem. ↓ | Error ↑ (%) | Tuned error ↑ (%) |
|---|---|---|---|---|---|
| 4.0× | - | 4.0× | 2.5× | +71.6 | +5.5 |
| - | 3.9× | 3.9× | 0.93× | +6.7 | +2.0 |
| 1.8× | 2.2× | 4.0× | 1.4× | +2.9 | **+1.6** |

Table 4: Acceleration of VGG-16, 4× theoretical speedup. First row is the proposed method, the second row is our reimplementation of linear asymmetric method of Zhang et al. [28], the third row is the combined method. Perforation is complementary to the acceleration method of Zhang *et al.*

4. While the decomposition method outperforms perforation, the combined method is better than both of the components.

## 5 Conclusion

We have presented PerforatedCNNs which exploit redundancy of intermediate representations of modern CNNs to reduce the evaluation time and memory consumption. Perforation requires only a minor modification of the convolution layer and obtains speedups close to theoretical ones on both CPU and GPU. Compared to the baselines, PerforatedCNNs achieve lower error, are more flexible and do not change the architecture of a CNN (number of parameters in the fully-connected layers and the size of the intermediate representations). Retaining the architecture allows to easily plug in PerforatedCNNs into the existing computer vision pipelines and only perform fine-tuning of the network, instead of complete retraining. Additionally, perforation can be combined with acceleration methods which exploit other types of network redundancy to achieve further speedups.

In future, we plan to explore the connection between PerforatedCNNs and visual attention by considering input-dependent perforation masks that can focus on the salient parts of the input. Unlike recent works on visual attention [1, 11, 20] which consider rectangular crops of an image, PerforatedCNNs can process non-rectangular and even disjoint salient parts of the image by choosing appropriate perforation masks in the convolutional layers.

**Acknowledgments.** We would like to thank Alexander Kirillov and Dmitry Kropotov for helpful discussions, and Yandex for providing computational resources for this project. This work was supported by RFBR project No. 15-31-20596 (mol-a-ved) and by Microsoft: Moscow State University Joint Research Center (RPD 1053945).

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
