[Supplementary Material · supplementary.pdf]



Figure 6: Comparison of with different interpolation strategies for perforated pixels. AlexNet network

## A   Interpolation strategy

In the paper, perforated values are interpolated using the value of the nearest neighbor. We compare this strategy to two alternatives: replacing with a constant zero and barycentric interpolation. For the second option, we perform Delaunay triangulation of the non-perforated points set. If a perforated point is in the interior of a triangle, then it is interpolated by a weighted sum of the values of the three vertices, with barycentric coordinates used as weights. Exterior perforated points are simply assigned the value of the nearest neighbor.

The results of comparison on AlexNet are presented on figure 6. We measure theoretical speedup (reduction of number of multiplications) to ignore the differences in implementations of the interpolation schemes. Replacing the missing values with zero is clearly not sufficient for successful acceleration of conv3 layer. Compared to the nearest neighbor, barycentric interpolation slightly improves results for pooling structure mask in conv2 and grid interpolation mask in the conv3 layer, but performs similarly or worse in other cases. Overall, nearest neighbor interpolation provides a good trade-off between complexity of the method (number of memory accesses per interpolated value) and the achieved error.

## B   Single layer results for NIN network

In section 4.1, we have considered single-layer acceleration of conv2 and conv3 layers of AlexNet. Here we present additional results for acceleration of the three non-$1 \times 1$ convolutional layers of NIN network. Each convolutional layer is followed by two $1 \times 1$ convolutions (which we treat as a part of non-linearity) and a pooling operation. Therefore, pooling structure mask applies to all layers. The results are presented on figure 7. We observe a similar pattern to the one observed in AlexNet conv2 and conv3 layers: grid and impact perforation masks perform best.

## C   Empirical and theoretical speedups

As noted in Denton et al. [6], achieving empirical speedups that are close to the theoretical ones (reduction of the number of multiplications) is quite complicated. We find that our method generally allows to do that, see table 5. For example, for theoretical speedup $4\times$, AlexNet conv2 empirical acceleration is $3.8\times$ for CPU and $3.5\times$ for GPU. The results are below the theoretical speedup in almost all cases due to the additional memory accesses required. The perforation mask type does

(a) conv1, CPU  (b) conv1, GPU  (c) conv2, CPU  (d) conv2, GPU

(e) conv3, CPU  (f) conv3, GPU

Figure 7: Acceleration of a single layer of CIFAR-10 NIN network for different mask types without fine-tuning. Values are averaged over 5 runs.

|  | NIN | | AlexNet | | VGG-16 | |
|---|---|---|---|---|---|---|
|  | CPU | GPU | CPU | GPU | CPU | GPU |
| conv1 | 4.4× | 2.7× | 3.2× | 2.7× | 2.5× | 2.2× |
| conv2 | 3.8× | 3.5× | 3.3× | 3.0× | 2.6× | 2.1× |
| conv3 | 3.7× | 3.3× | 4.1× | 3.7× | 3.2× | 2.5× |
| conv4 | - | - | 3.9× | 3.5× | 3.1× | 2.6× |
| conv5 | - | - | 3.6× | 3.4× | 3.5× | 2.8× |
| conv6 | - | - | - | - | 3.5× | 2.9× |
| conv7 | - | - | - | - | 3.4× | 2.9× |
| conv8 | - | - | - | - | 3.6× | 3.6× |
| conv9 | - | - | - | - | 3.6× | 3.7× |
| conv10 | - | - | - | - | 3.6× | 3.7× |
| conv11 | - | - | - | - | 3.7× | 3.6× |
| conv12 | - | - | - | - | 3.7× | 3.6× |
| conv13 | - | - | - | - | 3.8× | 3.6× |

Table 5: Per-layer empirical speedups for *uniform* perforation mask with $r = 0.75$. Theoretical speedup is $4\times$ in all cases. Results are averaged over 5 runs

not seem to affect the speedup. The difference between the empirical speedups on CPU and GPU highlights that it is important to choose per-layer perforation rates for the target device.

## D  Implementation details

A convolutional layer is typically applied to each image of the mini-batch sequentially. Fig. 8 shows the number of multiplications per second achieved by a quad-core Intel CPU and NVIDIA Geforce GTX 980 GPU on the bottleneck operation of evaluation of the convolutional layer: matrix multiplication of the data matrix $M$ by the kernel matrix $K$. We see that increasing the perforation rate reduces the efficiency of the operation, especially for GPU, which is as expected: GPUs work best for large inputs. Thus, for a fair comparison with the non-accelerated implementation, we stack $\lfloor \frac{1}{1-r} \rfloor$ images of the mini-batch, to match the size of the original data matrix. This requires a tensor transpose operation after the matrix multiplication, but we find that this operation is comparatively

Figure 8: The efficiency of matrix-by-matrix multiplication (measured in multiplications per second) of the data matrix $M$ by the kernel matrix $K$, for different perforation rates. AlexNet, the conv2 layer

fast. The same idea is used in MShadow library [2]. We also perform stacking of images for the baseline methods (resize, stride and fractional stride).