[Reviews · NeurIPS 2016]

Reviewer 1

Summary

The paper tries to speed-up CNN inference by simply not evaluating certain computational elements within a CNN architecture. Instead of evaluating them their outputs are interpolated by neighboring elements. The authors achieve a speedup of up to four times with the benefit of nearly directly using also pre-trained CNNs. However, fine-tuning is also necessary. Two strategies are proposed for the perforation: (1) with larger strides, (2) by learning the pixels with the largest influence on the loss.

Qualitative Assessment

Pros: (1) Clear, simple, and innovative idea (2) Well written paper. (3) Sufficient experiments. (4) The combination with [28] in 4.4. is very interesting. Cons: (1) "exploiting/removing the spatial redundancy" might be not the right wording, since it is unclear whether the information is really "redundant". (2) Interpolation with nearest neighboring is related to pooling strategies and should be discussed accordingly (3) A theoretical analysis would be interesting (not a real con). (4) Interpretation and discussion of the experiments should be improved. (5) The authors should evaluate their approach with ResNet/BatchNorm architectures. (6) The approach can not be combined with classical strategies that speed-up convolutions by FFTs (7) Is the benefit of not having to train networks from scratch really reasonable? If 1-day is needed for fine-tuning, a reduced version of ResNet/BatchNorm might be an alternative that could be learned from scratch as well also in reasonable time. (8) The authors should try more sophisticated interpolation strategies. (9) L106: why did the authors observe different slow-downs for CPU and GPU? Comments (1) L82: The authors mention a "first contribution" and I am missing the second one. (2) L111: The equation is just rearranged. (3) L116: uniform distribution "tends to cluster", why? (4) L126: This is a contradiction to the introduction: the impact method depends on the dataset. (5) L142-144: why not end-to-end? (6) Isn't it possible to learn r (Section 3.3)? (7) Table 2 mixes top-1 and top-5 error comparisions

Confidence in this Review

2-Confident (read it all; understood it all reasonably well)


Reviewer 2

Summary

The authors propose a method to accelarate the evaluation of CNNs by reducing the resolution of intermediate layers, interpolating the values of the network in between samples, and letting subsequent layers work as if the full resolution results were available. This is shown to be comparable to other network acceleration techniques that eliminate intermediate neurons (e.g. the group damage work of Lempitsky et al), while incurring a moderate loss in accuracy.

Qualitative Assessment

The main idea of the paper is to evaluate the network's output at fewer locations in the intermediate layers, interpolate the in-between values and perform the subsequent processing steps without really ever allocating memory or computation for the in-between values. There are two components here: one is the interpolation, which is technically straightforward (see e.g. the deconvolutional layers used in Shelhammer et al, CVPR 2015, for generalized variants). In my understanding the main contribution consists in avoiding the explicit storage of memory (l. 103-106: the interpolation is performed implicitly by masking the reads of the following pooling or convolutional layer). I understand the proposed technique as being very closely related to convolution with holes ('a trous' convolution), proposed in "Semantic Image Segmentation with Deep Convolutional Nets and Fully Connected CRFs", ICLR 2015, Liang-Chieh Chen, George Papandreou, Iasonas Kokkinos, Kevin Murphy, Alan L. Yuille, later on rebranded as 'dilated convolution' by Yu and Koltun, ICLR 2016. Namely, in the references above the convolution kernels become 'perforated' so as to increase the effective spatial resolution of a layer (ICLR 15) and/or the receptive field of a neuron (ICLR 16). The authors here are reinventing the same idea, but with the aim of reducing memory storage. Which is fine, but in my understanding very incremental. Given that a side-by-side comparison with the work of Lempitsky et al gives smaller accelerations and comparable, or larger reductions in error, I do not see what is the practical benefit.

Confidence in this Review

2-Confident (read it all; understood it all reasonably well)


Reviewer 3

Summary

This paper proposes a strategy for reducing computation cost of convolutional neural networks (CNNs) through partial evaluation and interpolation. As responses in a CNN feature map tend to be spatially correlated, this presents the opportunity to directly compute the values in the feature map at only a subset of all locations, and interpolate the values at the remaining locations. Interpolation is cheap, so this strategy trades compute time for a possible reduction in accuracy. The paper explores several choices for the set of locations at which to compute values (the perforation mask), including: uniform random, grid, pooling-based, weight-based, and impact based. Beyond random and grid patterns, these other alternatives attempt to choose between computation and interpolation based on minimizing the impact of interpolated values on the network's result. The impact perforation mask is computed by evaluating the approximate change in the CNN's loss function due to zeroing a position in some convolutional layer. Positions with smallest impact are then chosen for interpolation. The impact perforation mask for an entire network is found by alternating between computing impacts for one layer and then updating the network perforation and recomputing impacts. Experiments on common networks used in computer vision (NIN, AlexNet, VGG-16) reveal tradeoffs between speed and error for these perforation strategies. Additional experiments show the perforation technique can be combined with the acceleration technique of [28] to further improve speed.

Qualitative Assessment

Interpolation and partial evaluation of convolutional filters on a spatial domain are fairly basic ideas. Other recent work, [Yu and Kolten, ICLR 2016] and [Chen et al, ICLR 2015] also explore convolution operations with different input/output strides, albeit for slightly different purpose. This paper does make a potentially useful and novel contribution in terms of the proposed optimization procedure for finding network-specific perforation masks that minimize overall error. Experimental analysis is sufficiently thorough to give readers a good understanding of the speed vs accuracy tradeoffs for perforated convolution. These contributions overall would make this work a solid poster.

Confidence in this Review

2-Confident (read it all; understood it all reasonably well)


Reviewer 4

Summary

The paper proposes a novel acceleration method for convolutional neural networks based on modifying an existing network architecture to compute convolutions within a layer at only a subset of positions, with an interpolation method (typically nearest neighbor) used to fill in the values at the remaining positions. Experiments applying the acceleration method to NIN (CIFAR dataset) and AlexNet or VGG-16 (ImageNet dataset) demonstrate that speed ups of 2x to 4x of both CPU and GPU implementations can be obtained at only a moderate loss of accuracy after fine tuning. The proposed method is orthogonal to existing acceleration methods, and the authors confirm experimentally that it can be combined with existing acceleration methods for greater speedups.

Qualitative Assessment

The proposed method looks to be interesting and practical. The method for estimating impact appears novel. In Table 2, the results for [15] are given in terms of top-1 error while the other results are given in top-5 error (top-5 error should be clearly indicated in the table caption). This makes it difficult to compare them, and in particular to judge whether the speedup/error trade-off achieved by the proposed method used in combination with [15] is actually better than that achieved by [15] alone. More generally, the paper would benefit from a more thorough experimental comparison between the proposed method and existing acceleration methods, and combinations thereof. Table 3 indicates large differences in error between the GPU and CPU implementations. I would have expected the two implementations to give identical results, assuming identical parameters (I would not expect small differences in floating point results to affect the accuracy). The reason for this discrepancy should be clearly explained.

Confidence in this Review

2-Confident (read it all; understood it all reasonably well)


Reviewer 5

Summary

The paper describes a method to accelerate the evaluation of convolutional neural networks. The authors propose to skip computations in some of the positions of the convolutional layers (indicated with a binary "peforation" mask) and replace them with interpolated values. They empirically investigate the effects of different types of masks and perforation rates, and show that perforation can be combined with other methods to achieve additional speedups.

Qualitative Assessment

Overall I found this an interesting and well-written paper with immediate practical applications: it provides an "easy trick" to speed up existing convnets if one is prepared to suffer a small reduction in accuracy. The goal is clear and the proposed method well explained. Most figures and formulas are simple but helpful and correct. The paper also has some shortcomings, as explained below. - Technical quality The method description (section 3) is clear and seems correct. My main issue is that the empirical results from the experiments are not sufficiently analysed and discussed. Why does the proposed method work? What is the intuition or theory for why certain masks work better than others? Is there a more principled alternative to the greedy method for choosing the perforation configurations? These questions are not discussed. - Novelty/originality Although not groundbreaking, the method proposed in this paper seems sufficiently novel. - Potential impact or usefulness The paper provides a relatively straightforward way to speed up neural networks, which is certainly useful for deep learning practitioners. However, in contrast to some other acceleration methods, this approach requires a costly greedy search to find a good perforation configuration, and the adapted network needs retraining or finetuning. This might hinder large-scale adoption of this method in the deep learning community. - Clarity and presentation The paper is written clearly. Only the explanation of the grid perforation mask (line 118) is a bit too concise, and figure 2b does not help clarifying it. Perhaps the related work section could be extended a bit. Minor comments: - Figures 2 and 3 need better spacing - In figures 2c and 2d, it is strange that the pooling structure is a perfectly regular grid. Why is this? - No authors for citation 8 (line 278)

Confidence in this Review

2-Confident (read it all; understood it all reasonably well)


Reviewer 6

Summary

The goal of this paper is to reduce the evaluation time and memory size during evaluation of a CNN. The paper proposes by saving time by only evaluating a subset of convolutional features at selected spatial locations, determined by a “perforation mask”. The non-evaluated spatial locations are then filled in through nearest-neighbor interpolation. The paper proposes a few methods for determining this perforation mask. The paper evaluates the performance of this technique on recent CNN architectures: Network-in-network, AlexNet, and VGG-16.

Qualitative Assessment

This paper proposes a method to speed up the evaluation of CNNs. The method skips the evaluation of a subset of spatial locations in the image, and interpolates their values using nearest neighbor values. Methods for accelerating CNNs are of great interest to the community at this time. The approach of the method is well-described. I did have a few concerns with some of the results and parts of the method. (1) Figure 4(c)(d) shows the increase in error vs speedup factor for several methods. The impact line resulting in greater errors than all of the other methods in the conv3 layer is a surprising result, and some additional analysis would shed more light on the result to the reader. It may also indicate that the impact is not being computed with the correct metric. A feature which is not computed will be interpolated, rather than zeroed out, before evaluation in the next layer. Why then, is “impact” measured by estimating the effect of zeroing out the feature (feature magnitude times gradient), rather than just the effect of perturbing the feature (simply the gradient)? (2) The baselines all corrupt the CNN function. The “resize” operation changes the input scale of the image. The “stride” operation reduces the feature map resolution at the layer, so the subsequent layer is effectively performing an a trous/dilated convolution on the feature map at the original resolution, which means that it is evaluating a different function. I believe the true baseline is simply the perforated-CNN with zero-filling instead of interpolation. (3) Though there is no pooling layer directly after conv3 in AlexNet, the “pooling structure” method for perforation mask selection could reasonably be extended to conv4 and conv3, since layers conv3-conv5 share the same spatial locations. (4) For VGG, “pooling structure” will have a tie in all spatial locations. Since kernel&stride are equal in the network, no features are in overlapping regions. This should be mentioned, either in the page 3 line 123 or page 7 line 225. (5) The barycentric interpolation is likely to be of interest to the reader. I would recommend moving this material from the appendix into the paper, with an analysis of its classification performance, memory, and evaluation time.

Confidence in this Review

2-Confident (read it all; understood it all reasonably well)